# Effect of Yttrium-90 transarterial radioembolization in patients with non-surgical hepatocellular carcinoma: A systematic review and meta-analysis

**Simon Lemieux**[1,2☯]*, **Alex Buies**[1☯], **Alexis F. Turgeon**[3,4], **Julie Hallet**[5,6], **Gaétan Daigle**[7], **François Côté**[1], **Steeve Provencher**[2,8]

**1** Department of Radiology and Nuclear Medicine, Université Laval, Québec City, Québec, Canada, **2** Québec Hearth and Lung Institute Research Center, Université Laval, Québec City, Québec, Canada, **3** Division of Critical Care Medicine, Department of Anesthesiology and Critical Care Medicine, Université Laval, Québec City, Québec, Canada, **4** CHU de Québec–Université Laval Research Center, Population Health and Optimal Health Practices Research Unit, Trauma-Emergency-Critical Care Medicine, Québec City, Québec, Canada, **5** Division of General Surgery, Odette Cancer Centre, Sunnybrook Health Sciences Centre, Toronto, Ontario, Canada, **6** Department of Surgery, University of Toronto, Toronto, Ontario, Canada, **7** Faculty of Engineering Sciences, Department of Mathematics and Statistics, Université Laval, Québec City, Québec, Canada, **8** Division of Respirology, Department of Medicine, Université Laval, Québec City, Québec, Canada

☯ These authors contributed equally to this work.
* simon.lemieux.9@ulaval.ca

**Data Availability Statement:** All relevant data are within the manuscript and its Supporting Information files.

## Abstract

### Background

Recently, the use of Yttrium-90 transarterial radioembolization in non-surgical hepatocellular carcinoma was suggested but the evidence supporting its use is unclear.

### Methods

We searched Medline, Embase, Web of Science and Cochrane CENTRAL from inception up to April 14, 2020 for randomized controlled trials comparing Y90-TARE to standard of care in non-surgical HCC patients. Our primary outcome was overall survival (OS). Our secondary outcomes were progression-free survival, time to progression, disease control rate, grade ≥3 adverse events and rates of gastro-intestinal ulcers. Hazard ratios (HR) and risk ratios (RR) with random-effects model were used for our analyses. The risk of bias of the included studies was assessed using Cochrane's RoB 2 tool.

### Results

Of 1,604 citations identified, eight studies (1,439 patients) were included in our analysis. No improvement in overall survival were noted when Yttrium-90 transarterial radioembolization was compared to standard treatments (HR 0.99 [95% CI 0.81–1.21], 6 studies, $I^2$ = 77.6%). However, Yttrium-90 transarterial radioembolization was associated with fewer grade ≥3 adverse events (RR 0.64 [95% CI 0.45–0.92], 7 studies, $I^2$ = 66%). No difference was observed on other secondary outcomes.

**Funding:** The author(s) received no specific funding for this work.

**Competing interests:** The authors have declared that no competing interests exist.

## Discussion

In non-surgical HCC patients, Yttrium-90 transarterial radioembolization was not associated with significant effect on survival, progression-free survival, time to progression, disease control rate and the incidence of gastro-intestinal ulcers but was however associated with significantly lower rates of grade ≥3 adverse events. Further randomized controlled trials are warranted to better delineate optimal treatment.

## Introduction

Hepatocellular carcinoma (HCC) is the fourth leading cause of cancer-related deaths in the world, resulting in approximately 800,000 deaths globally annually [1]. It is typically diagnosed late in its course and the median survival following diagnosis ranges from 6 to 20 months [2]. At diagnosis, approximately only 30% of patients are eligible for curative treatments including surgery, mostly owing to extent of disease and patient comorbidities, including cirrhosis [2].

According to the Barcelona Clinic Liver Cancer (BCLC) staging system and guidelines, the standard treatment for intermediate HCC (BCLC stage B) is either conventional or drug-eluting beads transarterial chemoembolization (TACE) [3]. For advanced HCC (BCLC stage C), sorafenib recently became standard treatment after two trials documented benefits in overall survival (OS) [4,5]. However, although new treatments are available, the survival benefit is still not optimal and new alternatives are sought.

More recently, Y90-TARE was developed for the treatment of HCC and offers inherent advantages such as outpatient setting during a single treatment session [6,7]. However, despite significant amount of promising results derived from retrospective data [8,9], its use remains limited due to the uncertainty on its efficacy. Consequently, Y90-TARE is not considered a first-line treatment for HCC in recent guidelines [10–12]. However, Y90-TARE is offered to patients from early to terminal BCLC stage [13] in monotherapy or in combination.

Considering the potential benefit of Y90-TARE and the limited evidence supporting its use, we conducted a systematic review and meta-analysis to assess the efficacy and safety of Y90-TARE in non-surgical HCC patients.

## Methods

We conducted a systematic review in accordance with the framework from the Methodological Expectations of Cochrane Intervention Reviews [14]. We reported our work according to the Preferred Reporting Items for Systematic Reviews and Meta-Analyses guidelines (PRISMA) statement [15]. Patients or the public were not involved in the design, or conduct, or reporting or dissemination plans or our research. Our protocol was registered on Prospero (http://www.crd.york.ac.uk/PROSPERO, CRD42020179211).

## Study outcomes

Our primary outcome of interest was the overall survival (OS). Secondary outcomes included (1) time to radiological progression, defined whether as progression-free survival or time to progression at any site; (2) disease control rate, defined as the sum of complete response, partial response and stable disease; (3) severe/significant adverse events, defined as the proportion of patients who developed at least one grade ≥3 adverse event according to Common

Terminology Criteria for Adverse Events [16]; and (4) incidence of gastro-intestinal ulcers of any severity.

## Search strategy and selection criteria

We searched Medline, Embase, Web of Science and Cochrane CENTRAL from inception up to April 14, 2020. Our search strategy was developed in collaboration with an information specialist and is available online (S1 Text). Conference abstracts obtained through our search strategy were considered. We hand searched the National Institutes of Health clinicaltrials. gov. No language restriction was applied.

We included RCTs evaluating Y90-TARE (monotherapy or in combination) in non-surgical HCC patients compared to any treatment or intervention, placebo, sham-intervention, or no intervention. RCTs reporting at least one of our outcomes of interest were considered for inclusion.

Two reviewers (SL, AB) independently reviewed all trial titles and abstracts to determine eligibility. When pertinent, the full publication was assessed independently by the same reviewers to determine final inclusion. When duplicate populations from same trials were reported, the data from the largest cohort of patients was included in the final analysis. Discrepancies or uncertainties at any point were resolved by consensus. The agreement between the two reviewers was measured using the quadratic weighted κ statistic [17].

## Assessment of risk of bias

We assessed the risk of bias of the selected trials using Cochrane's revised tool for Risk of Bias in randomized trials (RoB 2) [18]. Two reviewers (SL, AB) independently completed the assessment of bias according to the "Template for completion" (available on riskofbias.org). The RoB 2 tool assesses five domains of bias: randomization process, deviations from the intended interventions, missing outcome data, measurement of the outcome, and selection of the reported results. We classified specifically all reported outcomes from each trial as low risk, some concerns, or high risk of bias.

## Quality of evidence

Two reviewers (SL, AB) independently applied the Grades of Recommendations Assessment, Development and Evaluation framework (GRADE) to assess external validity by evaluating the quality of evidence for all outcomes reported in the systematic review [19].

## Data extraction and statistical analyses

The same reviewers independently (SL, AB) extracted trial identification, name, year of publication, funding sources, disclosures, country of origin, number and location of participating centers, dates of conduction, inclusion/exclusion criteria, population, details about the interventions, including monotherapy or combination, and patient characteristics from included trials.

Trial data relevant to time-to-event outcomes (OS, progression-free survival, and time to progression) was analyzed using hazard ratios (HR) and extracted on a suggested data collection form [20]. We requested unpublished data from the authors when necessary. When the HR was not available, we extrapolated approximate individual patient data from published Kaplan-Meier curves with a digitizer software (DigitizeIt, Germany, available from digitizeit. de) to obtain the corresponding HR. The algorithm used is associated with a mean absolute error of 0.017 (95% CI 0.002–1.222), so that the true HR would be between 1.475–1.525 for an

extracted HR of 1.5, therefore representing no relevant systematic error [21]. For time-to-event outcomes, the (log)HR was used to summarize the results. The summary estimates were presented as a HR with its corresponding 95% CI using the inverse-variance method. For dichotomous outcomes, contingency (2x2) tables were constructed and number of events were recorded on the basis of the intervention received. The summary estimates were presented as a risk ratio (RR) with its corresponding 95% CI using the Mantel-Haenszel method. If the numerator cells contained values of zero, we added 1 to the numerator and denominator cells to calculate the RR. Statistical heterogeneity between trials was assessed according to $I^2$ with the following thresholds: (a) 0–40%: might not be important; (b) 30–60%: may represent moderate heterogeneity; (c) 50–90%: may represent substantial heterogeneity; and (d) 75–100%: considerable heterogeneity. We used the DerSimonian and Laird random-effects model [22] which accounts for within-trial and between-trial variability.

Statistical analyses were performed with Review Manager (version 5.3) and with R Software (version 4.0.0). We explored potential sources of heterogeneity with pre-specified subgroup analyses namely: (1) risk of bias; (2) type of comparator; (3) intervention in monotherapy or combination; (4) disease stage; (5) proportions of early and intermediate HCC (BCLC stages A and B); (6) proportions of advanced HCC (BCLC stage C); (7) portal vein invasion or tumor thrombosis; and (8) type of Y90 microsphere. We planned to visually assess publication bias using funnel plots, but the "rule of thumb" to include of a minimum of 10 trials was not fulfilled for any of the outcomes.

## Results

Of the 1,604 citations that were identified through our search strategy, eight RCTs [23–30] (1,439 patients, including one unpublished trial [26], for which study results were obtained via clinicaltrials.gov) met inclusion criteria (κ 0.93 [95% CI 0.78–1]). Flow chart and reasons for exclusion are presented in Fig 1. All trials compared Y90-TARE to another intervention including four trials (165 patients) comparing Y90-TARE to TACE [24,25,27,29], three trials (850 patients) comparing Y90-TARE to sorafenib [23,26,30], and one trial (424 patients) comparing the combination of Y90-TARE + sorafenib to sorafenib alone [28] (Table 1). Male patients represented 86% of the overall population. Of the included trials, 59%, 35.5%, and 5.5% of the patients had respectively advanced HCC, intermediate HCC, and early HCC. Y90-TARE was performed with resin microsphere in five trials [23,25,27,28,30] and glass microsphere in three trials [24,26,29].

### Risk of bias

For our primary outcome (OS), four trials were considered at low risk of bias (Fig 2, Table A in S1 Appendix) [23,24,29,30]. For two trials [25,27], the analytical plan was not described a priori and were therefore attributed as some concerns for risk bias. Two trials were considered at high risk of bias [26,28]. In one trial [26], seven participants were withdrawn after randomisation in the sorafenib group compared to one in the Y90-TARE which this trial also had missing outcome data. In the other trial [28], 47.2% of the participants did not receive the intervention they were allocated to or sustained major protocol deviations. A summary of the risk of bias assessments for the secondary outcomes is available in the online supplementary materials (S1 Appendix).

### Quality of the evidence

According to the GRADE framework, we rated the quality of evidence for overall survival (Table A in S2 Appendix). We rated the quality of evidence for the secondary outcomes as very low except for grade ≥3 adverse event as low (S2 Appendix).

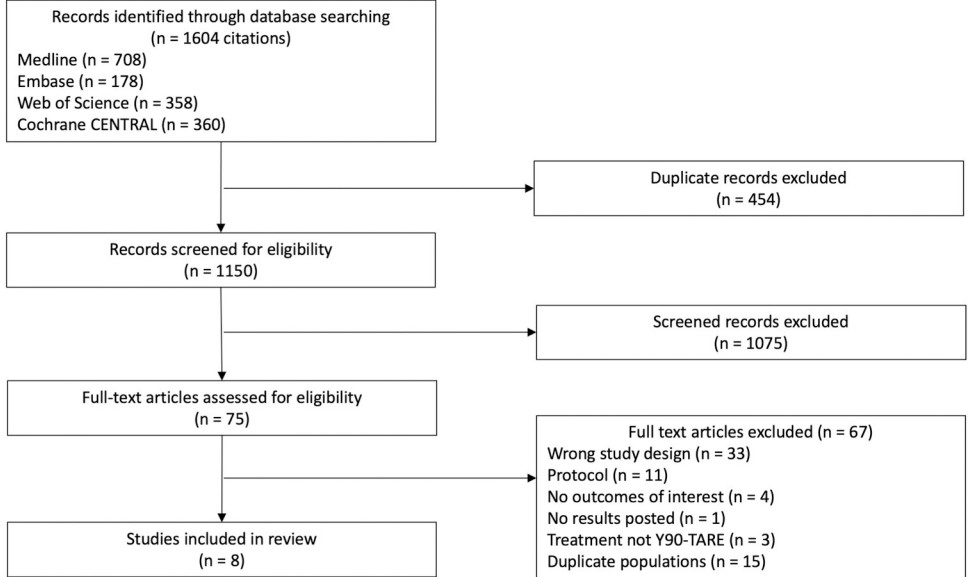

**Fig 1. PRISMA flow diagram.** The PRISMA flow diagram for the systematic review detailing the database searches, the number of citations screened, and the full texts retrieved.

## Primary outcome

The OS was reported in all eight trials. The corresponding HR was reported in three trials [23,28,30], obtained through correspondence with the author in one trial [24], and extrapolated from reconstructed approximate individual patient data from published Kaplan-Meier curves in two trials [27,29]. We could not use the data from two trials in our analyses as they either solely provided the median survival [26] or the survival rates at six and 12 months [25], respectively. Y90-TARE was not associated with differences in overall survival (HR 0.99 [95% CI 0.81–1.21], six trials, $I^2 = 77.6\%$, Fig 3) compared to standard of care. Our results were comparable in our subgroup analyses (Table 2).

## Secondary outcomes

**Time to radiological progression.** The time to radiological progression was reported as two different outcomes in the included trials: progression-free survival and/or time to progression. Progression-free survival was reported in four trials [23,25,27,30]. The corresponding HR was reported in two trials [23,30] and extrapolated from published Kaplan-Meier curves in one trial [27]. We could not use the data in our quantitative analyses in another trial [25] providing solely a median progression-free survival with its 95% CI. The time to progression was reported in five trials [23,24,27,29]. The corresponding HR were reported in two trials [23,29], obtained through author correspondence in one trial [24] and extrapolated using a previously described algorithm in one trial [27], but with only partial data. Y90-TARE yielded no differences in progression-free survival (Fig 3, Table A in S3 Appendix) and time to progression (Fig 3). Y90-TARE yielded a significantly longer time to progression in the glass microsphere subgroup (HR 0.23 [95% CI 0.12–0.45], two trials, $I^2 = 0\%$, Table B in S3 Appendix).

**Disease control rate.** Disease control rate was reported in five trials [23,25,26,29,30] and showed no difference between interventions (S1 Fig, Table C in S3 Appendix).

**Grade ≥3 adverse events and incidence of gastro-intestinal ulcers.** The number of patients who developed at least one grade ≥3 adverse event were reported in seven trials

**Table 1. Characteristics of included studies.**

| | Number of patients | Proportion of males (%) | BCLC stage n[a] (%) | | | Type of microsphere | Outcomes reported |
|---|---|---|---|---|---|---|---|
| | | | A | B | C | | |
| **Trials comparing Y90-TARE to TACE** | | | | | | | |
| Dhondt 2020 (TRACE trial) [24] | 68 | 59/68 (87%) | 9 (13%) | 59 (87%) | 0 | Glass | Primary: TTP |
| | | | | | | | Secondary: TLP, OS, overall response to therapy, toxicities and AEs, QoL, treatment-related costs |
| Kolligs 2015 (SIRTACE trial) [25] | 28 | 24/28 (86%) | 9 (32%) | 13 (46.5%) | 6 (21.5%) | Resin | Primary: HRQoL |
| | | | | | | | Secondary: PFS, survival, best objective tumor response, AEs |
| Pitton 2015 [27] | 24 | 18/24 (75%) | 1 (4%) | 23 (96%) | 0 | Resin | Primary: PFS |
| | | | | | | | Secondary: local tumor response, OS, TTP, nTTP |
| Salem 2016 (PREMIERE trial) [29] | 45 | 33/45 (73%) | 35 (78%) | 10 (22%) | 0 | Glass | Primary: TTP |
| | | | | | | | Secondary: OS, rate of response (DCR), and safety (AEs) |
| **Trials comparing Y90-TARE to sorafenib** | | | | | | | |
| Chow 2018 (SIRveNIB trial) [23] | 360 | 298/360 (83%) | 1 (0%) | 190 (53%) | 168 (47%) | Resin | Primary: OS |
| | | | | | | | Secondary: TRR, DCR, PFS, TTP, AEs, HRQoL |
| Mazzaferro 2019 (YES-P trial) [26] | 31 | 25/31 (81%) | 0 | 0 | 31 (100%) | Glass | Primary: OS |
| | | | | | | | Secondary: TTP, time to worsening portal vein thrombosis, time to symptomatic progression, tumor response, change from baseline in QoL, time to deterioration QoL, TEAE |
| Vilgrain 2017 (SARAH trial) [30] | 459 | 414/459 (90%) | 21 (4.5%) | 127 (27.5%) | 311 (68%) | Resin | Primary: OS |
| | | | | | | | Secondary: PFS, progression at any site, progression in the liver as the first event, tumor response, disease control, AEs, QoL |
| **Trials comparing Y90-TARE + sorafenib to sorafenib alone** | | | | | | | |
| Ricke 2019 (SORAMIC trial–palliative cohort) [28] | 424 | 358/419[b] (92%) | 9 (2%) | 124 (29%) | 284 (67%) | Resin | Primary: OS |
| | | | | | | | Secondary: AEs |

OS = overall survival. TRR = tumor response rate. DCR = disease control rate. PFS = progression-free survival. TTP = time to (tumor) progression. AEs = adverse events. HRQoL = health related quality of life. QoL = quality of life. TEAE = treatment emergent adverse events. TLP = time to local progression. nTTP = time to non-treatable progression.

[a]In the BCLC category, seven participants from Table 1 are missing from the SORAMIC trial (Ricke 2019) and one participant from Table 1 is missing from the SIRveNIB trial, due to unknown status.

[b]Five participants had unknown status of gender from the SORAMIC trial (Ricke 2019).

(1,245 patients) [23–26,28–30]. Y90-TARE was associated with significantly lower rates of grade ≥3 adverse event compared to standard treatment (RR 0.64 [95% CI 0.45–0.92], seven trials, $I^2$ = 66%, S1 Fig, Table D in S3 Appendix). This effect was associated with the use of sorafenib as a comparator, the absence of an active co-intervention and a balanced proportion of the different BCLC stages. No significant difference in the incidence of gastro-intestinal ulcers was noted (four trials [23,25,28,30]) (S1 Fig, Table E in S3 Appendix).

A summary of findings table is available online (S1 Table).

## Discussion

In our systematic review with meta-analyses, we observed that Y90-TARE yields no effect in the overall survival, progression-free survival, time to progression, disease control rate and incidence of gastro-intestinal ulcers when compared to standard of care treatment in non-surgical patients with HCC. Our results were consistent in all other subgroup analyses, except for the use of glass microsphere that seemed to improve the time to progression. We observed that

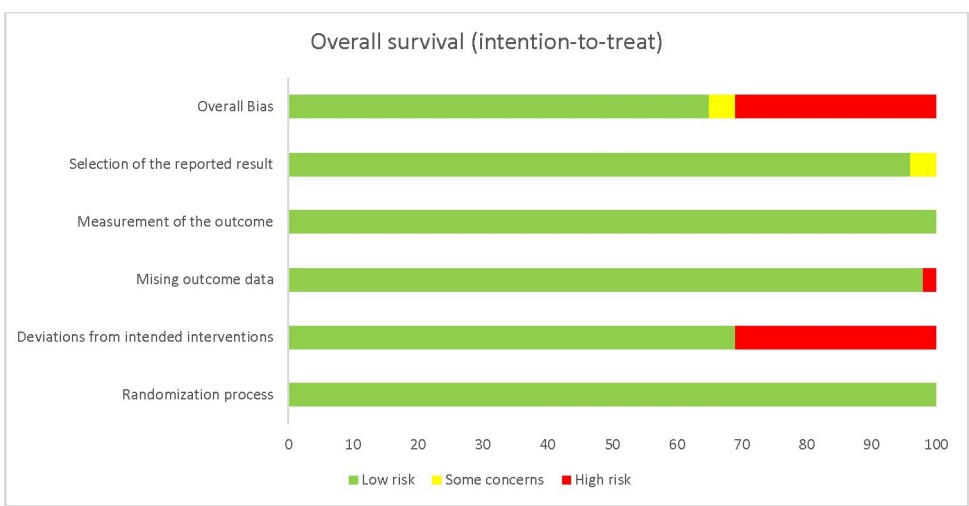

**Fig 2. Risk of bias of the primary outcome, overall survival.** Weighted graph showing the risk of bias according to each domain. Green = low risk. Yellow = some concerns. Red = high risk.

grade ≥3 adverse events were significantly less frequent with Y90-TARE compared to the standard of care. However, subgroup analyses showed that this effect was explained by the use of sorafenib as a comparator, the use of an active co-intervention and a balanced proportion of the different BCLC stages.

The results of our systematic review for the overall survival are aligned with those of five systematic reviews showing no effect with the use of Y90-TARE [8,31–34], but are in contradiction with three other systematic reviews [9,35,36]. Our work is however more exhaustive and not limited to one specific comparator in the context of absence of a unique standard of care. Previous systematic reviews suggesting a survival benefit with the use of Y90-TARE [9,35,36] had major limitations in their design including the consideration of non-randomised controlled studies. Differing from other systematic reviews [8,31–33], our work examined both progression-free survival and time to progression, which are outcomes used as primary endpoints in some randomized controlled trials in HCC. Our results suggesting that the use of glass microsphere Y90-TARE may possibly be associated with longer time to progression as compared to standard of care treatment differ from prior meta-analyses [8,9,31–36]. Previous studies comparing the resin and glass microsphere to deliver the treatment showed conflicting results [37,38]. Our observation was derived from a subgroup analysis of only two trials and we cannot exclude a type 1 error.

## Strength and limitations

Our study has several strengths. Our study population is exhaustive and, as compared to other systematic reviews, we expressed time-to-event outcomes using HRs [39]. Median survival times was the most reported measurement of the included trials in our meta-analysis, but is not an optimal pooled estimate for survival data [40]. However, underreporting of relevant information concerning survival analysis, notably the HR, was observed in retrieved trials in our systematic review [41]. We managed this limitation in two of the included trials [27,29] by reconstructing individual patient data from published Kaplan-Meier curves using a proposed algorithm [21], enabling us to present a cumulative pooled HRs of time-to-event outcomes, which represents the most up-to-date and relevant effect estimates. Also, the comprehensiveness of the search strategy renders unlikely the omission of pertinent trials.

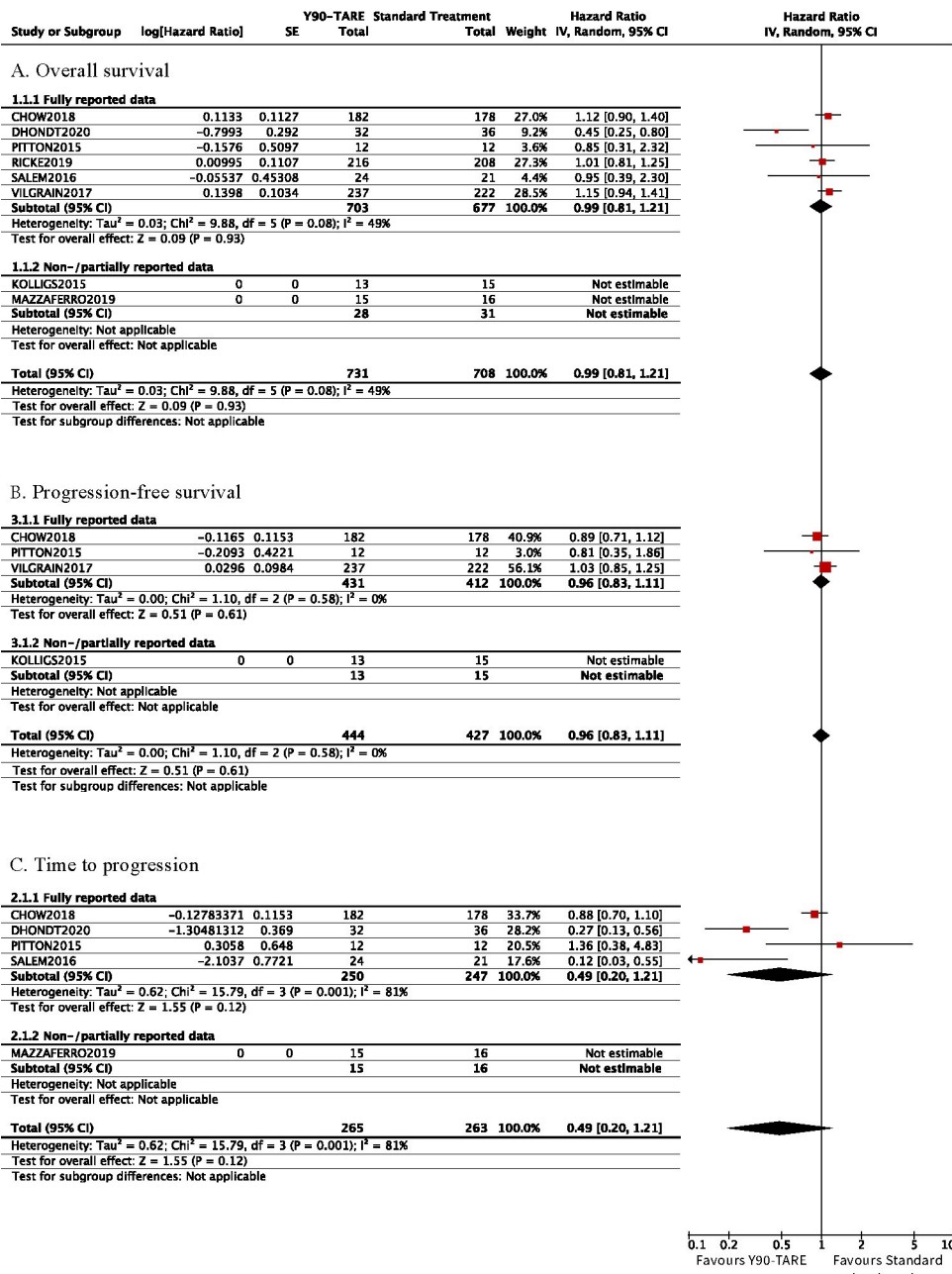

**Fig 3. Forest plot of time-to-event outcomes.** Cumulative (log)HR estimates with their 95% confidence intervals in the random-effects model for (A) overall survival, (B) progression-free survival, and (C) time to progression.

Our systematic review also has limitations. First, despite being more exhaustive than previous work, we included a limited number of trials and most of the largest trials were designed to compare Y90-TARE to sorafenib. Moreover, neither the HR nor a Kaplan-Meier curve were provided in two small trials, precluding the extrapolation of missing data on survival and thus reducing our sample size. Most subgroup analyses were also not very robust considering the limited number of trials. For our primary outcome, two of the trials were considered at high risk of bias and two with some concerns on the risk of bias. In addition, for all secondary outcomes except adverse events, most trials were at high risk of bias or of some concerns. The

**Table 2. Predefined subgroup analyses for the primary outcome.**

| | Number of trials[a] | Population | | Random-effects model | | Heterogeneity |
|---|---|---|---|---|---|---|
| | | Y90-TARE | Standard treatment | Pooled HR | 95% CI | I$^2$ (%) |
| **Risk of bias (according to RoB 2)** | | | | | | |
| Low | 4[23, 24, 29, 30] | 475 | 457 | 0.95 | 0.69–1.30 | 69% |
| Some concerns | 1[27] | 12 | 12 | 0.85 | 0.31–2.32 | N/A |
| High | 1[28] | 216 | 208 | 1.01 | 0.81–1.25 | N/A |
| **Type of comparator** | | | | | | |
| TACE | 3[24, 27, 29] | 68 | 69 | 0.63 | 0.38–1.04 | 19% |
| Sorafenib | 3[23, 28, 30] | 635 | 608 | 1.09 | 0.97–1.24 | 0% |
| **Co-treatment** | | | | | | |
| Systemic | 1[28] | 216 | 208 | 1.01 | 0.81–1.25 | N/A |
| None | 5[23, 24, 27, 29] | 487 | 469 | 0.95 | 0.71–1.26 | 59% |
| **Disease stage** | | | | | | |
| Mostly BCLC A-B | 4[23, 24, 27, 29] | 250 | 247 | 0.81 | 0.49–1.35 | 65% |
| Mostly BCLC C | 2[28, 30] | 453 | 430 | 1.08 | 0.93–1.26 | 0% |
| **Proportion of BCLC stages A and B** | | | | | | |
| <33% | 2[28, 30] | 453 | 430 | 1.08 | 0.93–1.26 | 0% |
| 33–66% | 1[23] | 182 | 178 | 1.12 | 0.90–1.40 | N/A |
| >66% | 3[24, 27, 29] | 68 | 69 | 0.63 | 0.38–1.04 | 19% |
| **Proportion of BCLC stage C** | | | | | | |
| <33% | 3[24, 27, 29] | 68 | 69 | 0.63 | 0.38–1.04 | 19% |
| 33–66% | 1[23] | 182 | 178 | 1.12 | 0.90–1.40 | N/A |
| >66% | 2[28, 30] | 453 | 430 | 1.08 | 0.93–1.26 | 0% |
| **Portal vein invasion or tumor thrombosis** | | | | | | |
| Majority | 1[30] | 237 | 222 | 1.15 | 0.94–1.41 | N/A |
| Minority | 2[23, 28] | 398 | 386 | 1.06 | 0.91–1.24 | 0% |
| None | 3[24, 27, 29] | 68 | 69 | 0.63 | 0.38–1.04 | 19% |
| **Type of Y90 microsphere** | | | | | | |
| Resin | 4[23, 27, 28, 30] | 647 | 620 | 1.09 | 0.96–1.23 | 0% |
| Glass | 2[24, 29] | 56 | 57 | 0.60 | 0.30–1.23 | 48% |

Y90-TARE = Yttrium-90 transarterial radioembolization. HR = hazard ratio. CI = confidence interval. N/A = not applicable.

[a]The SIRTACE (Kolligs 2015) and YES-P (Mazzaferro 2019) trials had partially reported data and are therefore not included in the effect estimate and subgroup analyses for OS.

quality of the evidence was thus affected by these important limitations. Also, reporting bias could not be excluded as the few numbers of included RCTs precluded the reliable assessment of the funnel plot. Finally, as combination therapies are increasingly being used in contemporary HCC treatments, our meta-analysis did not aim to evaluate optimal sequencing of therapy.

## Conclusion

In non-surgical HCC patients, we did not observe a significant improvement with the use of Y90-TARE on overall survival, progression-free survival, time to progression, disease control rate and gastro-intestinal ulcer rates. Y90-TARE was associated with significantly lower rates of grade ≥3 adverse events which may be related to the use of sorafenib as a comparator, the absence of an active co-intervention and a balanced proportion of the different BCLC stages. However, the small number of trials and limited sample size may explain this later finding.

Well-designed RCTs evaluating the effect of Y90-TARE compared to standard of care on survival are warranted to better delineate its role in the treatment of non-surgical HCC considering the quality of the current evidence.

## Supporting information

**S1 Checklist.**
(DOC)

**S1 Fig. Forest plot of the cumulative risk ratio of dichotomous outcomes.**
(PDF)

**S1 Table. Summary of findings.**
(DOCX)

**S1 Appendix. Assessment of the risk of bias of included trials for all outcomes.**
(DOCX)

**S2 Appendix. Quality of evidence according to the GRADE framework.**
(DOCX)

**S3 Appendix. Subgroup analyses of the secondary outcomes.**
(DOCX)

**S1 Text. Full search strategy.**
(DOCX)

## Author Contributions

**Conceptualization:** Simon Lemieux, Alex Buies, Alexis F. Turgeon, Julie Hallet, François Côté, Steeve Provencher.

**Data curation:** Simon Lemieux, Alex Buies, Gaétan Daigle, François Côté, Steeve Provencher.

**Formal analysis:** Simon Lemieux, Alex Buies, Alexis F. Turgeon, Gaétan Daigle, Steeve Provencher.

**Investigation:** Simon Lemieux, Alex Buies, Alexis F. Turgeon, François Côté.

**Methodology:** Simon Lemieux, Alex Buies, Alexis F. Turgeon, Julie Hallet, Gaétan Daigle, Steeve Provencher.

**Project administration:** Simon Lemieux.

**Software:** Gaétan Daigle.

**Supervision:** Simon Lemieux, Alex Buies, Alexis F. Turgeon, Julie Hallet, Gaétan Daigle, François Côté, Steeve Provencher.

**Validation:** Simon Lemieux, Alex Buies, Alexis F. Turgeon, Gaétan Daigle, François Côté, Steeve Provencher.

**Visualization:** Simon Lemieux, Alex Buies, Alexis F. Turgeon, Steeve Provencher.

**Writing – original draft:** Simon Lemieux, Alex Buies, Steeve Provencher.

**Writing – review & editing:** Simon Lemieux, Alex Buies, Alexis F. Turgeon, Julie Hallet, Gaétan Daigle, François Côté, Steeve Provencher.

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
