## [Decision Letter · Decision Letter 0]

10 Dec 2020

PONE-D-20-34186

Effect of Yttrium-90 transarterial radioembolization in patients with non-surgical hepatocellular carcinoma: a systematic review and meta-analysis

PLOS ONE

Dear Dr. Lemieux,

Thank you for submitting your manuscript to PLOS ONE. After careful consideration, we feel that it has merit but does not fully meet PLOS ONE’s publication criteria as it currently stands. Therefore, we invite you to submit a revised version of the manuscript that addresses the points raised during the review process.

We look forward to receiving your revised manuscript.

Kind regards,

Yi-Hsiang Huang, M.D., Ph.D.

Academic Editor

PLOS ONE

Journal Requirements:

2. Please amend your list of authors on the manuscript to ensure that each author is linked to an affiliation. Authors’ affiliations should reflect the institution where the work was done (if authors moved subsequently, you can also list the new affiliation stating “current affiliation:….” as necessary).

Reviewers' comments:

Reviewer's Responses to Questions

**Comments to the Author**

1. Is the manuscript technically sound, and do the data support the conclusions?

Reviewer #1: Yes

Reviewer #2: Partly

2. Has the statistical analysis been performed appropriately and rigorously? 

Reviewer #1: N/A

Reviewer #2: Yes

3. Have the authors made all data underlying the findings in their manuscript fully available?

Reviewer #1: Yes

Reviewer #2: Yes

4. Is the manuscript presented in an intelligible fashion and written in standard English?

Reviewer #1: Yes

Reviewer #2: Yes

5. Review Comments to the Author

Reviewer #1: The authors conducted a systematic review and meta-analysis study to evaluate the outcome of Yttrium-90 transarterial radioembolization in patients with non-surgical

hepatocellular carcinoma. There are several comments:

I. Strengths:

A. This study was conducted under an exhaustive search and unpublished data from the reference authors were requested.

B. All the enrolled studies were published RCTs and is different from prior meta-analysis studies which included cohort studies.

II. Weakness:

The enrolled studies were categorized into Y90-TARE vs. cTACE and Y90-TARE vs. Sorafenib initially, but in data analysis, all the studies were mixed and were divided as Y90-TARE vs. standard of care. Most subgroup analyses were also not very robust considering the limited number of trials

Reviewer #2: In this systemic review with meta-analysis, the authors observed that Y90-TARE does not have significant clinical benefits for tumor control or survival in patients with non-surgical HCC.

Major comments

1. Although the methods and statistics are correct in this study, major limitation still could not be ignored in the enrolled studies because 2 of 8 trials with a high risk of bias and 2 trials with concerns of bias in analysis of overall survival. In addition, most trials were at high risk of bias or of concerns in analysis of secondary outcomes, including progression-free survival, time to progression, disease control rate, etc. Such limitations would have great impact on the results in this study. The authors would at least eliminate two trials of high risk of bias in analysis of primary outcome.

2. Subgroup analysis according to the presence of co-treatment with systemic therapy was conducted for the rates of gastrointestinal ulcers. The authors still have better to perform similar subgroup analysis in other outcomes.

Minor comments

1. There are many clinical evidences real-world data to support the new treatments of advanced or unresectable HCC. Therefore, the authors have better to revise the introduction “although new treatments are available, the survival benefit is still not optimal and new alternatives are sought”.

2. Citation should be added on the each trial in the table 1.

3. Y90-TARE seems to yield a significantly longer time to progression in the glass microsphere subgroup. The authors have to discuss this finding well.

6. PLOS authors have the option to publish the peer review history of their article (what does this mean?). If published, this will include your full peer review and any attached files.

Reviewer #1: No

Reviewer #2: No

---

## [Author Response · Author response to Decision Letter 0]

22 Jan 2021

Comments from the editors:

We have revised our manuscript according to PLOS ONE’s style requirements. We have also reviewed all file naming.

2. Please amend your list of authors on the manuscript to ensure that each author is linked to an affiliation. Authors’ affiliations should reflect the institution where the work was done (if authors moved subsequently, you can also list the new affiliation stating “current affiliation:….” as necessary).

The list of authors has been revised and modifications have been made to ensure that each author is linked to an affiliation.

Response to reviewer #1 comments:

Strengths:

A. This study was conducted under an exhaustive search and unpublished data from the reference authors were requested.

B. All the enrolled studies were published RCTs and is different from prior meta-analysis studies which included cohort studies.

Thank you for noting this.

Weakness:

The enrolled studies were categorized into Y90-TARE vs. cTACE and Y90-TARE vs. Sorafenib initially, but in data analysis, all the studies were mixed and were divided as Y90-TARE vs. standard of care. Most subgroup analyses were also not very robust considering the limited number of trials

We thank the reviewer for his comment. The main purpose of a meta-analysis is to synthesize treatment effects considering all available data. By pooling data from different trials, the power to detect a differential effect is increased. Trials included in a systematic review frequently present some degree of clinical heterogeneity. This recommended approach improves the ability to answer important research questions and understand the modifiers of important treatment effects, such as type of comparator treatment. We however planned subgroup analyses a priori to evaluate whether the type of comparator could explain the findings. 

Response to reviewer #2 comments:

Major comments

1. Although the methods and statistics are correct in this study, major limitation still could not be ignored in the enrolled studies because 2 of 8 trials with a high risk of bias and 2 trials with concerns of bias in analysis of overall survival. In addition, most trials were at high risk of bias or of concerns in analysis of secondary outcomes, including progression-free survival, time to progression, disease control rate, etc. Such limitations would have great impact on the results in this study. The authors would at least eliminate two trials of high risk of bias in analysis of primary outcome.

We thank the reviewer for the thorough assessment of our work. As highlighted in our risk of bias assessment section, some of the included trials were at high risk of bias. However, rather than excluding trials, we explored the potential effect of the risk of bias in our subgroup analyses, as suggested by the Cochrane Methodology of systematic reviews and meta-analysis.[1] Subgroup analyses are in such situation more informative than sensitivity analyses where removing analyses would only inform of the remaining subgroup. 

2. Subgroup analysis according to the presence of co-treatment with systemic therapy was conducted for the rates of gastrointestinal ulcers. The authors still have better to perform similar subgroup analysis in other outcomes.

We performed subgroup analyses for each of our outcomes when possible to do so. Some could however not be conducted due to the absence of trials in each subgroup category. Progression-free survival, time to progression, and disease control rate were outcomes which only included Y90-TARE without co-treatment. We added specifications regarding the subgroup analyses that could not be conducted in S3 Appendix (under Tables A, B, and C).

Minor comments

1. There are many clinical evidences real-world data to support the new treatments of advanced or unresectable HCC. Therefore, the authors have better to revise the introduction “although new treatments are available, the survival benefit is still not optimal and new alternatives are sought”.

Thank you for your comment. Although many new treatments showing promising results are available, the quality of the evidence is still not optimal to be very assertive on the matter. We therefore prefer to keep the same phrasing.

2. Citation should be added on each trial in the table 1.

The change has been made to Table 1.

3. Y90-TARE seems to yield a significantly longer time to progression in the glass microsphere subgroup. The authors have to discuss this finding well.

Thank you for your comment. We modified the first paragraph of the discussion to highlight this finding and we also put this result in context with prior knowledge in the second paragraph of the discussion.

 

REFERENCES:

1. Boutron I, Page MJ, Higgins JPT, Altman DG, Lundh A, Hróbjartsson A. Chapter 7: Considering bias and conflicts of interest among the included studies. In: Higgins JPT, Thomas J, Chandler J, Cumpston M, Li T, Page MJ, Welch VA (editors). Cochrane Handbook for Systematic Reviews of Interventions version 6.1 (updated September 2020). Cochrane, 2020. Available from www.training.cochrane.org/handbook.

---

## [Decision Letter · Decision Letter 1]

17 Feb 2021

Effect of Yttrium-90 transarterial radioembolization in patients with non-surgical hepatocellular carcinoma: a systematic review and meta-analysis

PONE-D-20-34186R1

Dear Dr. Lemieux,

We’re pleased to inform you that your manuscript has been judged scientifically suitable for publication and will be formally accepted for publication once it meets all outstanding technical requirements.

Kind regards,

Yi-Hsiang Huang, M.D., Ph.D.

Academic Editor

PLOS ONE

Additional Editor Comments (optional):

Reviewers' comments:

Reviewer's Responses to Questions

**Comments to the Author**

1. If the authors have adequately addressed your comments raised in a previous round of review and you feel that this manuscript is now acceptable for publication, you may indicate that here to bypass the “Comments to the Author” section, enter your conflict of interest statement in the “Confidential to Editor” section, and submit your "Accept" recommendation.

Reviewer #1: All comments have been addressed

2. Is the manuscript technically sound, and do the data support the conclusions?

Reviewer #1: Yes

3. Has the statistical analysis been performed appropriately and rigorously? 

Reviewer #1: Yes

4. Have the authors made all data underlying the findings in their manuscript fully available?

Reviewer #1: Yes

5. Is the manuscript presented in an intelligible fashion and written in standard English?

Reviewer #1: Yes

6. Review Comments to the Author

Reviewer #1: The authors conducted more detailed subgroup analyses and explanation to support the strengths of this study. I have no further comments.

7. PLOS authors have the option to publish the peer review history of their article (what does this mean?). If published, this will include your full peer review and any attached files.

Reviewer #1: No

---

## [Editor Report · Acceptance letter]

22 Feb 2021

PONE-D-20-34186R1 

Effect of Yttrium-90 transarterial radioembolization in patients with non-surgical hepatocellular carcinoma: a systematic review and meta-analysis 

Dear Dr. Lemieux:

I'm pleased to inform you that your manuscript has been deemed suitable for publication in PLOS ONE. Congratulations! Your manuscript is now with our production department. 

Kind regards, 

on behalf of

Professor Yi-Hsiang Huang 

Academic Editor

PLOS ONE